# Premotor cortex hemodynamic responses primarily reflect perceptual rather than specific motor aspects of decision making

Jeff Boucher[1,2]*, Shihab Shamma[1,3], Yves Boubenec[1]*

**1** Laboratoire des systèmes perceptifs, Département d'études cognitives, École Normale Supérieure, PSL University, CNRS, Paris, France, **2** Ear Institute, University College London, London, United Kingdom, **3** Neural Systems Laboratory, University of Maryland, College Park, Maryland, United States of America

\* jeffrey.boucher@ucl.ac.uk (JB); boubenec@ens.fr (YB)

## Abstract

Decisions are driven by perception, but also by non-perceptual factors. It remains an open question, however, whether frontal brain regions involved in perceptual decision-making tend to uniquely reflect the perception of an animal, or the final choice of action driven by perceptual and non-perceptual factors. Using functional ultrasound imaging (fUSI), we investigated how the premotor cortex (PMC) in ferrets represents stimuli in a Go/NoGo auditory categorization task, varying the difficulty in order to manipulate the rates of perceptual errors relative to non-perceptual errors. We found that on Easy error trials, where error in perception was less likely, PMC activity was similar to correct answers for the same stimulus category. In contrast, on Difficult error trials, PMC activity more closely reflected the choice the animal made, being similar to correct answers for the opposite category. These results together are consistent with PMC activity reflecting the reward-predictive perceptual category. Perceptual errors could be refined further by assessing licking patterns, but licking patterns alone did not explain the effect. This study advances our understanding of the functional role of the frontal cortex in decision-making, suggesting that the PMC integrates sensory inputs to guide behavior based on perceptual information, rather than motivational information.

## 1 Introduction

Perceptual decision-making is the process by which sensory information is analyzed and converted into an action. In order to study this phenomenon, scientists often train their model animals on specifically designed tasks, with the intention that the animal reports their perception faithfully. A basic feature of such perceptual tasks is that the closer the perceptual objects to discriminate are, the more difficult the task and the more errors the participants will make [1]. These *stimulus-dependent* errors are often

**Data availability statement:** Column-formatted data used for all major analyses is located at https://doi.org/10.5281/zenodo.19354970, via Zenodo. Also included are motion energy data and piezo data, as well as the full codebase used for the project and the specific data underlying figures 2B, 2C, 2D, 2E, 3C, 3D, 3E, 4B, 4D, 4E, 4F, 4G, and 4H. Raw imaging data and video data is too large to be uploaded but will be stored on the lab's NAS at Laboratoire des Systèmes Perceptifs, and will be made available on request.

**Funding:** This work was supported by ANR-17-EURE-0017 and ANR-10-IDEX-0001-02 to YB, JB, and SS, CDSN doctoral fellowship from École Normale Supérieure to JB, ERC 787836-NEUME and ANR-19-CE37-0016 to SAS, and Institut Universitaire de France (IUF), ANR-JCJC-DynaMiC, and SESAME funding PSL-fUS to YB. The funders had no role in study design, data collection and analysis, decision to publish, or preparation of the manuscript.

**Competing interests:** The authors have declared that no competing interests exist.

**Abbreviations:** AM, Amplitude Modulated; dCBV, change in cortical blood volume; FA, False Alarm; fUSI, functional ultrasound imaging; HRF, hemodynamic response function; ITI, inter-trial interval; OFC, orbital frontal cortex; PFC, prefrontal cortex; PMC, premotor cortex.

interpreted as evidence of the participants' misperception, evidence that they have committed a *perceptual* error.

However, incorrect choices can also be due to *non-perceptual* (or *stimulus-independent*) errors. In humans, a common additional cause of such errors is a lapse in attention [2]. In addition to lapses of attention, a human may also intentionally make a wrong choice in order to explore options, rather than continuing to exploit a reliable pattern of behavior, if they expect that the optimal action might change [3]. Much more rarely (with the stronger effects in a neuro-psychological context with lesions to the ventral prefrontal cortex [PFC]) [4–6], a human will demonstrate a strong bias for a previously rewarding action and be unable to prevent themselves from choosing it, even if sensory evidence obviously points in the other direction. In each of these cases, the action the participant took may not faithfully represent their true perception.

Non-human animals express each of these non-perceptual errors as well. Several studies have demonstrated that non-human animals show biases towards certain actions [7,8] or explorative behavior [9]. In particular, in an appetitive Go/NoGo paradigm, an animal is expected to have a strong bias for the choice which grants them a reward [10], in a manner analogous to human patients. In these situations, the animals' actions may not accurately report their perceptions.

In the ferret, the frontal cortex has recently begun to be investigated under auditory and visual perceptual tasks [11–14]. It has been shown that cells in both the PFC and the premotor cortex (PMC) respond in a categorical manner to task stimuli, responding to the stimulus's meaning within the task rather than its physical parameters [12]. It remains an open question, however, whether the responses in this region tend to reflect the perception of the ferret or the final choice of action of the ferret.

Analysis of error trials might allow for a closing of this gap, particularly if we consider that not all error trials are equal: an error trial with a stimulus on the slope (central part) of the psychometric curve (a *Difficult* stimulus) should be relatively more likely to involve a *perceptual* error (Fig 1A); in this case, the stimulus would be internally classified as the incorrect category. For stimuli in the saturated regions (flat parts) of the psychometric curve (*Easy* stimuli), errors will relatively more often be *non-perceptual*, rather than miscategorizations of the stimulus. As a consequence, it should be more probable in Easy error trials (relative to Difficult error trials) that a stimulus would be correctly categorized internally, despite the incorrect choice.

Using this logic, we can disentangle the following: do PMC categorical responses reflect the category internally assigned to the stimulus, or do they reflect the enacted choice? If the choice is primarily what matters, all error trials involving the same action should show similar patterns of brain activity, irrespective of difficulty. If, however, the internal category of the stimulus is the primary driver of activity in PMC, activity during non-perceptual errors (more likely in errors on Easy trials) should look like that of the correct answers, and brain activity during perceptual errors (more likely in errors on Difficult trials) should look more like that of the opposite stimulus category (reflecting the miscategorization). Such logic could be applied to this

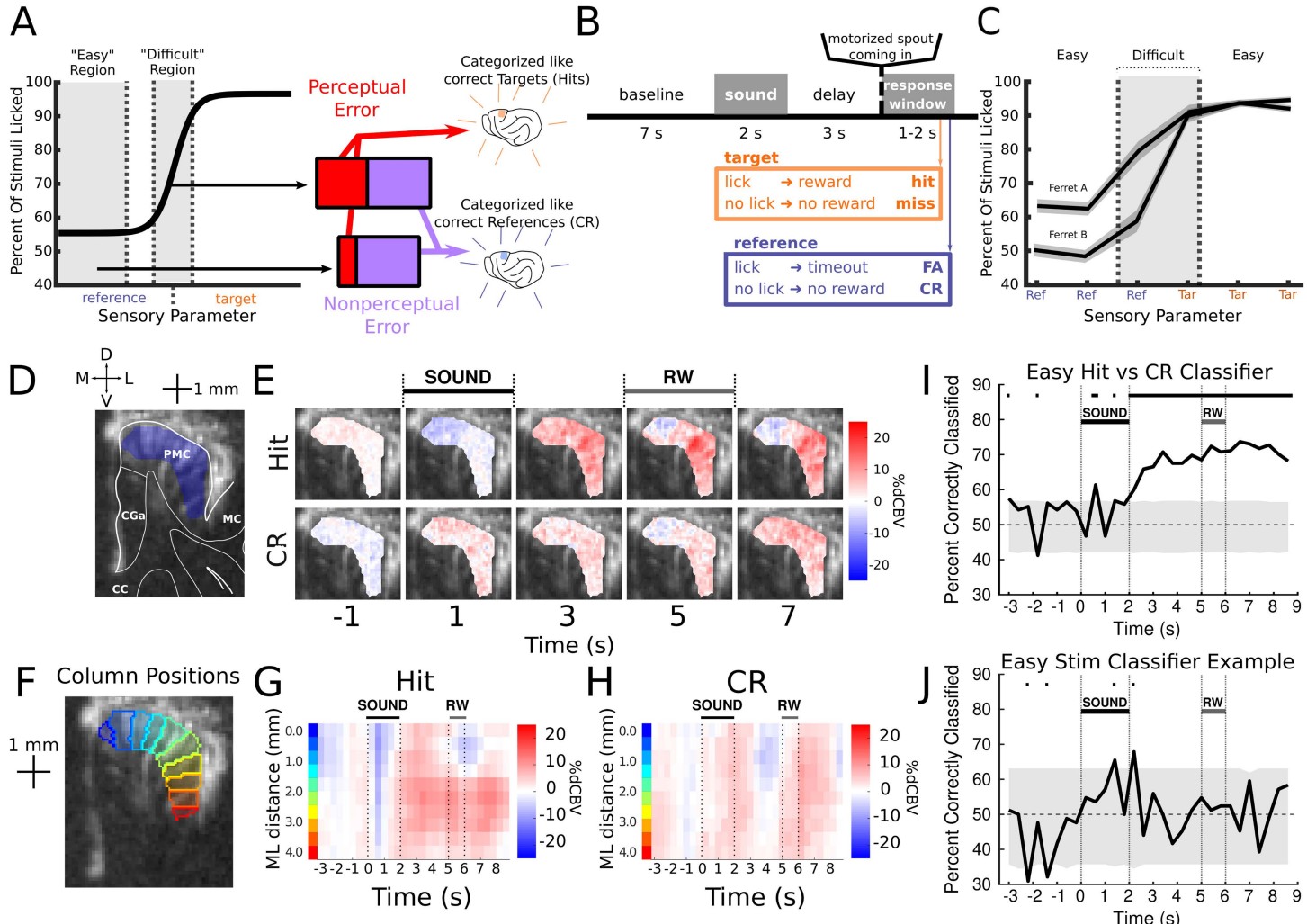

**Fig 1. Task and data preparation. A**: Cartoon illustrating the important parts of a psychometric function, the expected ratios of perceptual/non-perceptual errors in each region, and how we expect each type of error to map onto neural activity. For this binary Go/NoGo task, the two decisions depend on the value of the sensory parameter, and the correct decision can be defined at a specific boundary. At a far enough distance from this boundary (with the specific distance depending on the sensory parameter being studied) performance level becomes constant; errors may still be made, but they are independent of the stimulus and potentially independent of perception. Close to the boundary, errors increase; this is understood to be due to perceptions becoming more ambiguous in this region. We expect perceptual errors to show activity similar to hits, and non-perceptual errors to show activity similar to CRs. Because the ratio of perceptual errors should be higher in the Difficult region, we expect the mean pattern of activity to be closer to the Hit pattern, relative to errors in the Easy region. **B**: Cartoon describing trial structure. **C**: Measured psychometric curves for ferrets A and B. **D**: Cartoon showing how to read the anatomy of a coronal fUSI image. **E**: Example stimulus-evoked responses plotted per voxel. Voxel activity varies mainly along the surface of gray matter, and varies little across different depths. Only selected time bins are shown due to space concerns, but the full timecourse can be seen in panels G and H. **F**: Positions of the "columns" for the example session and slice. See Materials and methods for how they were constructed. **G**: Mean stimulus-evoked response for Hit trials, using the mean values of the columns in F. The important variations shown in panel E can be seen here, in addition to other timeframes. **H**: Same as G, but for CR trials. **I**: The percent correctly classified for the Easy Hit-CR classifiers for each time frame. All sessions recorded at this slice were included to train the classifiers, including the example session of the prior figures. Classifiers were trained using the column means (visualized in panels G and H). The shaded area contains the 2.5% to 97.5% quantiles of the shuffled percents correctly classified. **J**: Same as I, but comparing the two easy reference stimuli for the frequency rule. This example pair was chosen arbitrarily; all other pairs show similarly negative results, supporting the interpretation that responses are categorical.

question in any animal model and brain region with categorical activity, and to our knowledge has not yet been applied in any other instance.

To address this question, we recorded hemodynamic responses in ferret frontal cortex during behavior with functional ultrasound imaging (fUSI), a wide-scale hemodynamic imaging technique [15,16]. Using fUSI, we scanned task-evoked neural responses in PMC while ferrets performed a delayed-report Go/NoGo task (Fig 1B), in which the ferrets needed to lick (Go) after "Target" stimuli and refrain from licking (NoGo) on "Reference" trials. We were able to find distinct patterns of hemodynamic responses between correct Target (Hit) and Reference (Correct Rejections/CR) trials in PMC. Ferrets were trained to discriminate stimuli that were perceptually far from one another (Easy) as well as close to one another (Difficult). Hemodynamic response patterns for Easy and Difficult CR were similar, confirming that, when correct, Easy and Difficult trials showed categorical responses in PMC. Meanwhile, False Alarm (FA) errors (i.e., incorrect "Go" responses to Reference stimuli) were also similar to CR trials, but only in Easy trials, in line with the "stimulus category" hypothesis presented above. The Difficult FA trials were instead consistently closer to Hit trials than the Easy FA trials were, again in favor of the stimulus category hypothesis. This suggests that in ferret PMC, hemodynamic activity follows the pattern expected if it primarily responded to the internal sensory category, rather than the final decision made.

## 2 Results

### 2.1 Ferrets perform tasks which span the psychometric curve

Ferrets were trained on a Go/No-Go delayed categorization task under appetitive reinforcement. Water-deprived ferrets classified 2-second Amplitude Modulated (AM) pure tones into two categories: "Target" (ferret must lick) or "Reference" (ferret must suppress licking) based on carrier frequency or AM rate. Specific properties of these stimuli are elaborated in the Materials and methods section and S1A Fig, but are not critical to our presentation of the conclusions. Stimuli were followed by a 3-second delay during which the water spout was out of reach (Fig 1B). Licks during the subsequent response window were rewarded with water in Target Hit trials and punished with a timeout in Reference FA trials.

The task was designed to sample key response accuracies along a sigmoid-shaped psychometric curve, with stimuli represented in both its saturated and sloped regions (Figs 1A and S1). Performance reached a d' above 1 for both ferrets for the "Easy" stimuli (ferret A: $d' = 1.2$, Hit rate $0.94 \pm 0.007$, $n = 1,120$ trials, FA rate $0.63 \pm 0.014$, $n = 1,182$ trials; ferret B: $d' = 1.48$, Hit rate $0.93 \pm 0.007$, $n = 1,353$ trials, FA rate $0.49 \pm 0.013$, $n = 1,409$ trials; Fig 1C). Performance was saturated (flat/unchanging) for Easy stimuli, as can be demonstrated by the indistinguishability of performance between them (for References, ferret A $p = 0.78$, $n = 590$ Easy Reference stim 1, 592 Easy Reference stim 2; ferret B $p = 0.49$, $n = 711$ Easy Reference stim 1, 698 Easy Reference stim 2, two-tailed two sample $z$-test, for Targets, ferret A $p = 0.50$, $n = 555$ Easy Target stim 1, 565 Easy Target stim 2; ferret B $p = 0.22$, $n = 672$ Easy Target stim 1, 681 Easy Target stim 2, two-tailed two sample z-test).

Meanwhile, ferrets performed worse but decently on the difficult stimuli, as intended (ferret A: $d' = 0.51$, Difficult FA rate $0.79 \pm 0.03$, $n = 193$ trials, Difficult Hit rate $0.91 \pm 0.02$, $n = 183$; ferret B: $d' = 1.1$, Difficult FA rate $0.59 \pm 0.03$, $n = 235$ trials, Difficult Hit Rate $0.90 \pm 0.02$, $n = 228$; Fig 1C). FA rates for Reference stimuli were significantly higher for Difficult stimuli compared to Easy stimuli (Fig 1C), suggesting they were in the sloped part of the psychometric curve (ferret A $p < 10^{-5}$, $n = 193$ Difficult Reference trials, 1,182 Easy Reference trials; ferret B $p = 0.004$, $n = 235$ Difficult Reference trials, 1,409 Easy Reference trials, one-tailed two sample z-test). Hit rates for Difficult Target stimuli were similar to but slightly lower than those for Easy Target stimuli (ferret A: Hit rate 0.91, ferret B: Hit rate 0.90) suggesting that ferrets may not have actually found these stimuli difficult (ferret A $p = 0.06$, $n = 183$ Difficult Target trials, 1,120 Easy Target trials; ferret B $p = 0.06$, $n = 228$ Difficult Target trials, 1,353 Easy Target trials, one-tailed two sample z-test). An inability to find a significant effect here may have had to do with the ferrets' low decision criterion; this also led to a lack of difficult miss trials to analyze (ferret A $n = 17$ Difficult Misses, ferret B $n = 22$ Difficult Misses). For these reasons, analysis of difficulty effects will focus on Reference stimuli and FA trials.

## 2.2 Premotor cortex hemodynamics distinguish between Hit and Correct Rejection trials during active behavior

We first established differences in large-scale hemodynamic responses during correct Target (Hit) and Reference (Correct Rejection; CR) trials. We recorded blood volume changes (%dCBV) in PMC of two behaving ferrets using fUSI (Fig 1D and 1E). Recordings were performed repeatedly at specific positions along the anterior-posterior axis in PMC (34 sessions for ferret A; 37 sessions for ferret B), mapped to the ferret brain atlas [17] through anatomical landmarks (S2 Fig). For clarity, examples throughout this paper focus on one PMC position from ferret A (17 sessions); results generalized across positions and animals (see S3 and S4 Figs).

We found distinct PMC hemodynamic responses between Hit and CR trials. Target sounds evoked a transient suppression at around 1 s (Fig 1E), followed by a sustained CBV increase throughout the delay. This pattern remained clear after pooling voxels into anatomically defined 0.5 mm-wide columns, allowing alignment across sessions (Fig 1F–1H; Materials and methods and S5 Fig).

To further quantify neural representations of Hit and CR trials, we performed linear decoding to identify time-dependent optimal linear classification axes (Fig 1I). Models were trained on each time bin and slice separately, allowing for tracking of decoding accuracy over the trial timecourse. Hit versus CR decoding accuracy became significant 2 seconds after stimulus onset and plateaued approximately 1 second later. This latency aligns well with the expected timescale of fUSI hemodynamics; the hemodynamic response function (HRF) for fUSI has previously been reported as peaking 2 s post-neural activity onset ([18,19]). This indicates consistency with previously reported single-neuron PMC responses [12,13]. Additionally, stimuli from within the same category could not be decoded from each other (Fig 1J), confirming categorical PMC responses. Mean neural responses to each stimulus, for each region and ferret, can additionally be seen in S6 Fig.

## 2.3 PMC activity reflects internal rather than expressed categories on False Alarm trials

We investigated whether ferret PMC activity correlated more with internally-classified categories or expressed behavior by examining FAs during Easy and Difficult trials. As laid out in the introduction, FAs in the steep region of the psychometric function (Difficult trials) likely reflect genuine perceptual miscategorization, whereas those in the saturated region (Easy trials) more commonly reflect non-perceptual errors. However, FAs in either case involve a lick action. If evoked activity during Difficult and Easy FAs were similar, this may imply that the decision to lick was driving the response. We hypothesized, however, that PMC activity would primarily reflect the ferret's internally classified category for the stimulus. If this were the case, PMC categorical responses should represent the *Target* category in a Difficult *Reference* FA trial, as *perceptual* errors are more likely. However, in Easy *Reference* FA trials, PMC activity would look more similar to the *Reference* stimulus, as *non-perceptual* errors should be likely. Note that, across difficulty levels, the total error rate reflects a mixture of perceptual and non-perceptual errors, with the proportion of perceptual errors increasing with difficulty but never reaching 100%. Even so, this shift in the underlying mixture is detectable in the mean PMC response, leading to different average activity patterns for Easy versus Difficult FAs (Fig 1A).

To test this hypothesis, we projected Easy and Difficult FA trials onto Hit-CR classifiers trained for each slice and time point (Figs 1I and 2A). This procedure reduced multicolumn PMC responses to a single dimension encoding similarity to Hit or CR patterns. Easy FA trials showed greater similarity to CR trials despite involving a lick decision (Figs 2B, S3B, S3F, S3J, S3N, and S3R). In contrast, Difficult FA trials, likely reflecting misperceptions, projected closer to Hits (Figs 2C, S3B, S3F, S3J, S3N, and S3R). This difference is unlikely due to stimulus identity, as Difficult and Easy Reference trials aligned closely during Correct Rejections (Figs 2D, S3B, S3F, S3J, S3N, and S3R).

Across recording positions and ferrets, Difficult FA projections were significantly closer to Hits than Easy FA trials (Fig 2E; ferret A $p < 10^{-5}$, ferret B $p = 0.007$) and Difficult CR trials (ferret A $p < 10^{-5}$, ferret B $p < 10^{-4}$). For data aggregation, each slice was normalized so mean Hit projections equaled 1 and mean CR projections equaled −1 (Fig 2E). Altogether, these results strongly indicate PMC activity reflects internal stimulus classification rather than expressed behavioral decisions.

 

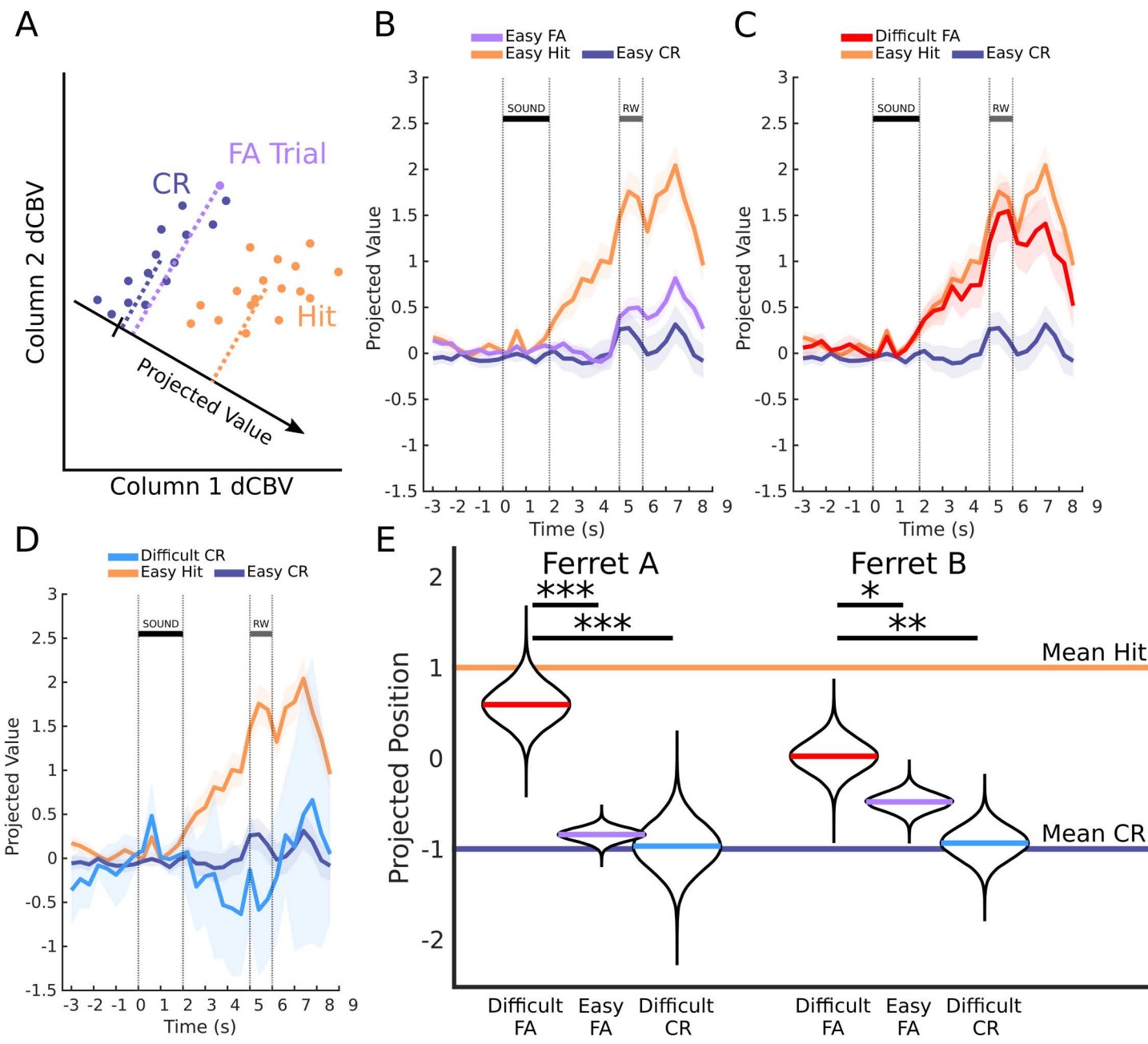

**Fig 2. False alarms project similarly to correct rejections for easy but not for difficult trials. A**: Schematic of projections onto an optimal classification axis. The linear classifier will determine the direction in multiple dimensions (two are plotted here) for which two categories (Hit and CR) are classified. The positions of a Projected Value along this axis can then inform the extent to which the classifier would consider a trial to be a CR trial or a Hit trial. Arbitrary new trials, as long as they use comparable units (i.e., the same columns), can be meaningfully projected on this axis. **B**: Mean projections of PMC hemodynamic activity in Easy FA trials onto the example slice's optimal axis. The mean projections of test-set Hit and CR trials are also plotted for reference. Shaded areas are 2 SEM. The HRF has an approximately 2-second delay, so interpretations of the timings should be adjusted to this. Note that the mean projection for Easy FA is aligned to Easy CR (same stimulus, different decisions), particularly in the seconds just following sound onset. **C**: Same as B, but for Difficult FA trials. Unlike Easy FA trials, Difficult FAs are aligned with the Easy Hit trials. **D**: Same as C, but for Difficult CR trials. The FA and CR projections for Difficult trials (C and D) do not look similar; Difficult CR trials look like Easy CR trials. **E**: Projections of the data aggregated across slices and across time frames from 3.4 to 5.4 seconds post-stimulus-onset (see Materials and methods). All categories are colored the same as the prior panels; the mean Hit projection is enforced to be 1 and the mean CR is enforced to be −1 (see Materials and methods), so they are plotted as horizontal lines. The bootstrapped means of Difficult FA trials for both ferrets are significantly separable from Easy FA trials and from Difficult CR trials. The data underlying figures B, C, D, and E can be found at https://doi.org/10.5281/zenodo.19354970.

## 2.4 A characteristic licking behavior further refines perceptual errors in Difficult trials

We previously established that FAs can arise from multiple causes, with Difficult-trial errors likely stemming from perceptual miscategorization. To further refine this analysis, we examined ferrets' licking behaviors as potential indicators of confidence and reward expectation. During FA trials, ferrets often exhibited "Persistent" licking, continuously licking despite receiving no water (Fig 3A and 3B), suggesting anticipation of reward or licking bias. In other FA trials, animals showed "Restrained" licking, licking just once (Fig 3A and 3B. We hypothesized that Persistent licking in Difficult trials indicates perceptual miscategorization more reliably than Restrained licking, potentially facilitating differentiation between perceptual and non-perceptual errors.

We split FA trials into Persistent and Restrained categories based on extended licking observed in the late response window (see Materials and methods). We projected these trial types separately onto previously determined Hit-CR classification axes (Figs 3C, S3C, S3G, S3K, S3O, and S3S). Persistent Lick trials consistently projected closer to Hits than Restrained Lick trials across slices and ferrets, when those trials were Difficult, and particularly in ferret B (Fig 3E; ferret A $p = 0.033$, ferret B $p = 0.0008$). This suggests that further refinement of FA trials to increase the proportion of perceptual errors may be possible.

To further validate the binarization into Persistent versus Restrained licking, we quantified how often trials were classified as Persistent as a function of trial type and difficulty (S7 Fig). For FAs, the fraction of Persistent-lick trials was higher for Difficult than for Easy stimuli in both animals, consistent with Persistent licking being more likely (though not exclusively) on trials where sensory evidence is weaker. For Hits, the proportion of Persistent trials was higher for Easy than Difficult stimuli in Ferret A, with the same but non-significant trend in Ferret B, suggesting that stronger sensory evidence can also promote more vigorous licking on correct detections. These distributions support the idea that our binary classification captures a behaviorally meaningful dimension of licking.

Persistent licking naturally involved greater motor activity. To control for the possibility that motor activity alone influenced these differences, we performed a similar analysis on Easy FA trials, which lack perceptual ambiguity. We hypothesized Persistent licks in Easy trials mainly reflected licking bias rather than perceptual errors, predicting minimal differences between Persistent and Restrained licking in these cases. Note here that Persistent and Restrained licks look qualitatively similar across these categories (Fig 3A and 3B).

We tested this by projecting Easy FA trials categorized as Persistent or Restrained onto the Hit-CR classifiers (Fig 3D). Unlike Difficult trials, we observed no significant projection difference between Persistent and Restrained lick Easy trials (ferret A $p = 0.14$, ferret B $p = 0.21$; Fig 3E). Thus, licking behavior specifically refines perceptual from non-perceptual errors primarily in Difficult FA trials.

## 2.5 Lick-related activity cannot explain differences in PMC activity between Easy and Difficult trials

Ferrets were not rewarded or punished for licking outside the response window, but sometimes licked early in response to stimuli, before spout arrival (Fig 4A). This behavior depended on stimulus type and the ferret's eventual response (Fig 4B, top), though substantial variability existed across trials (Fig 4B, bottom). To account for potential effects of licking behavior on neural activity, we removed lick-related activity using ridge regression models incorporating regressors based on ferrets' lick rates at multiple delays (1.2–2.4 s preceding the time frame) and video motion energy (see Materials and methods). We subtracted the regression contributions from PMC hemodynamic activity and re-ran the full analysis, retraining Hit-CR classifiers and projecting FA data onto these updated classifiers. Classifiers still robustly discriminated Hit from CR trials in the critical time window (3.4–5.4 s) for all ferrets and slices (Figs 4C, S4A, S4E, S4I, S4M, and S4Q), except briefly post-spout-arrival for two slices (Figs 4C and S4M).

Qualitatively, the results after regression subtraction remained consistent with prior analyses (Figs 2 and 3). Specifically, the main finding—that Difficult FA activity projected closer to Hits compared to Easy FA trials (ferret A $p < 10^{-5}$, ferret B $p = 0.04$) and Difficult CR trials (ferret A $p = 0.0256$, ferret B $p = 0.02$)—remained robust (Fig 4D–4G). Similarly, among

**Fig 3. A characteristic licking behavior further refines perceptual errors in Difficult trials. A**: Raster plot of licks for Difficult FA trials. Trials are sorted into "Persistent" and "Restrained". **B**: Same as A but for a randomly-selected subset of Easy-FA trials. Note the qualitative similarity of Persistent and Restrained licks across FAs between panels A and B. **C**: Mean projections of PMC neural activity in Difficult FA trials (in the manner of Fig 2) grouped into "Persistent" licks and "Restrained" licks. When the ferret restrains itself following an initial FA in the reward window (dotted line), their Difficult FA activity projects closer to the Easy CR mean. When they make additional attempts, the projection is closer to the Hit mean. **D**: Same as C, but with Easy trials instead of Difficult trials. In the case of Easy trials, the "Persistent lick" behavior is less likely to reflect a misperception. Consistent with that idea, we see that Persistent licks and Restrained licks project to the same position for Easy FA trials, despite being distinguished using the same method as for Difficult trials. **E**: Projections of the data aggregated across slices and time frames (same format and method as Fig 2E). For both ferrets, Persistent lick trials project significantly closer to Hit trials than Restrained lick trials. For both ferrets, there is no distinguishing between Persistent and Restrained lick trials for the Easy FA trials. The data underlying figures C, D, and E can be found at https://doi.org/10.5281/zenodo.19354970.

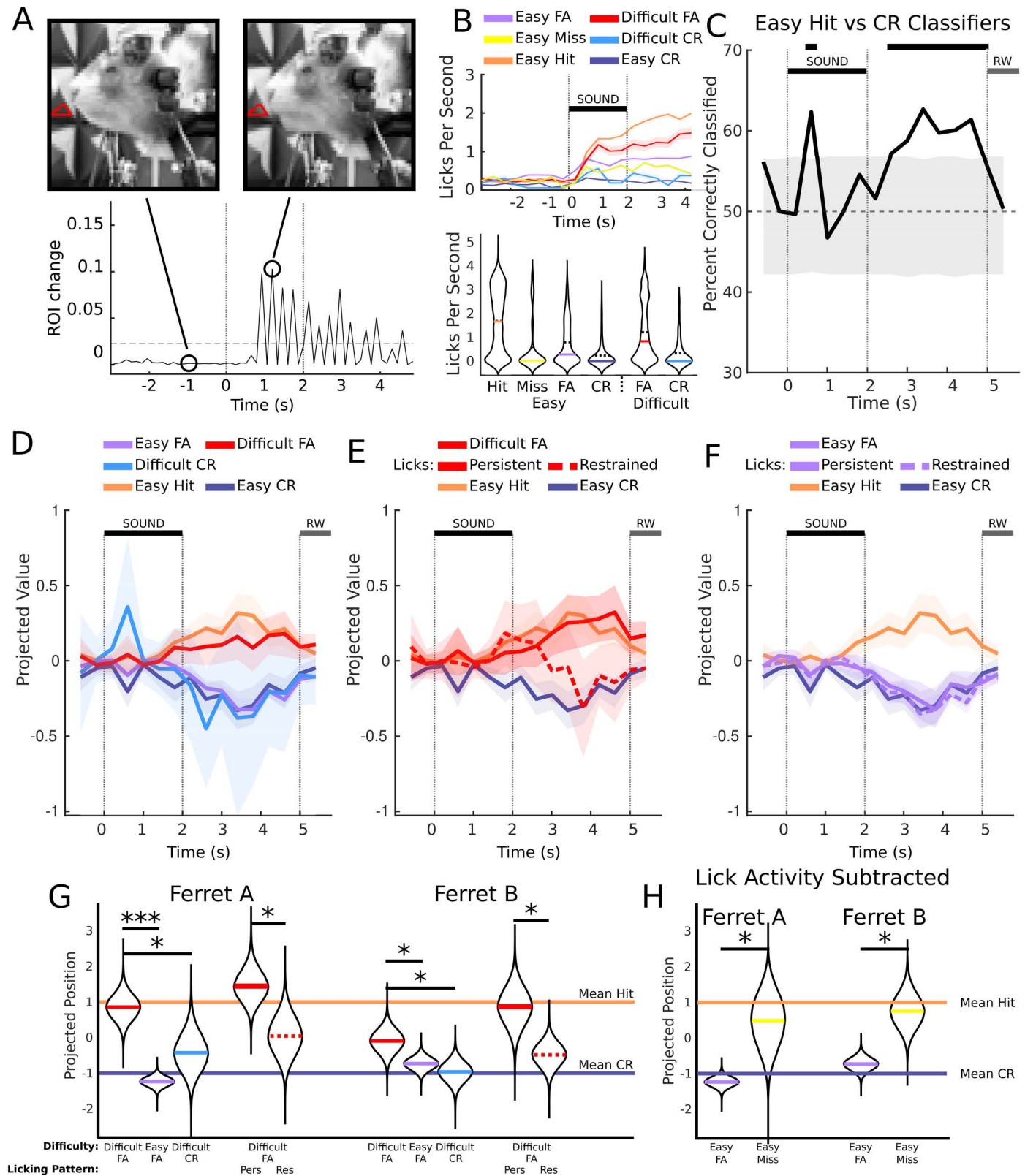

**Fig 4. Pre-spout licking cannot explain differences in PMC activity between Easy and Difficult trials. A**: Illustration of the video lick detection method. We drew ROIs where licks might occur, and tracked changes in them (see Materials and methods). **B**: Top: Average lick rates for different

response types and difficulties (ferret A). Shaded areas are the standard error. Bottom: Violin plots showing average lick rate in each trial over the period from 1 to 4.6 seconds. The dotted line is the mean, the colored is the median. **C**: Easy Hit-CR decoding accuracy from PMC neural activity (as in Fig 1I) after subtracting the movement-related contribution. The absolute magnitude of the classification has been reduced, suggesting that lick rate contributed to the classification ability. However, classification is still significant in the post-stimulus period, suggesting differentiating activity unrelated to this. **D**: Mean projections of PMC neural activity for Difficult FA, Easy FA, and Difficult CR (as in Fig 2B, 2C, and 2D) using the regression-subtraction classifiers. Difficult FA trials still project close to the mean Hit projections, and Easy FA and Difficult CR trials still project to the mean Easy CR projection. **E**: Persistent/Restrained Lick trial projection plot for Difficult FA trials (as in Fig 3A). Restrained licks still project close to Easy CR trials, despite being Difficult FA trials. **F**: Persistent/Restrained Lick trial projection plot for Easy FA trials (as in Fig 3B). The licking pattern on spout arrival does not affect the projected value of Easy FA trials. **G**: Projections of the data aggregated across slices, replicating prior results after lick subtraction. Persistent/Restrained Lick trials for Easy FA are omitted for space, but would be aligned with the Easy FA means. **H**: Projections of Easy FA and Easy Miss trials following lick subtraction. After accounting for lick-related activity, Easy Misses project quite significantly more toward the mean Hit activity for both ferrets, relative to Easy FAs. The data underlying figures B, D, E, F, G, and H can be found at https://doi.org/10.5281/zenodo.19354970.

Difficult FA trials, Persistent Lick trials projected significantly closer to Hits compared to Restrained Lick trials (ferret A $p = 0.02$, ferret B $p = 0.02$), while Easy FA trials showed no difference between Persistent and Restrained licking (ferret A $p = 0.21$, ferret B $p = 0.34$). Therefore, while licking behavior may influence classifier shape, it does not alter the observed distinctions between perceptual and non-perceptual errors in PMC activity.

In addition, performing the regression subtraction revealed that Easy Miss trials project more closely to Hit trials than to CR trials relative to Easy FA trials, as would be predicted if the activity reflected the reward-predictive categorization (ferret A $p = 0.01$, ferret B $p = 0.001$) (Fig 4H). Without accounting for licks, the effect was insignificant (ferret A $p = 0.07$, ferret B $p = 0.07$), potentially as a result of the large difference in mean lick rate between Hit and Miss trials (Fig 4B).

## 3 Discussion

Perceptual decision-making involves both perceptual and non-perceptual factors. Disentangling these per trial is challenging, but theoretical models like signal detection theory allow experimental control over the likelihood of errors from each factor. Here, we leveraged differences in perceptual error likelihood to demonstrate that ferret PMC hemodynamic responses primarily reflect perceptual rather than motor aspects of decision-making.

### 3.1 fUS imaging of PMC in behaving ferrets and implications at the neuronal level

We used fUSI to explore PMC encoding of sensory evidence and categorization during decision-making. Our study demonstrates the utility of fUSI in behaving animals, previously shown primarily in primates [20]. fUSI allowed us to obtain mesoscale, columnar maps of task-related hemodynamics across multiple anterior-posterior positions in and around PMC, with high spatial resolution and stable anatomical registration across many days compared to other hemodynamic-based techniques. This wide coverage is difficult to achieve with standard chronic electrode arrays in ferret frontal cortex, where simultaneously sampling multiple neighboring columns and re-targeting exactly the same sites across weeks, remains challenging. This allowed us to reliably track the same regions across days anatomically, allowing the aggregation of rarer error trials which made this study feasible. However, fUSI measures neural activity indirectly through blood flow changes rather than directly from neuronal firing. Thus, our findings reflect integrated neural population responses rather than individual neuron activity. In line with previous work showing that neuronal response properties in ferret frontal cortex are varied and not strictly anatomically localized [12], it is likely that the region of PMC we studied contains mixed-selectivity neurons jointly sensitive to sensory category, action, and reward expectation, as previously reported in rodents and primates [21,22]. Our data demonstrate that at the hemodynamic level PMC activity tracks the animal's internal auditory category, but they cannot specify the relative contribution of these different neuronal subpopulations. A natural next step is therefore to use targeted electrophysiology guided by these fUSI maps. Based on the present results and existing work in frontal cortex, we would expect single-unit and population recordings in ferret PMC to reveal neurons whose firing (i) tracks the internal auditory category even when report and category diverge (e.g., Easy FAs), and (ii) more

closely reflects reward anticipation or motor preparation. Our fUSI data identify specific PMC columns and time windows where such signals are likely to be strongest. Additionally, such a study could make use of a two-alternative forced choice paradigm, in which both sensory categories are rewarded to better dissociate sensory category signals from pure reward expectation, albeit potentially at the cost of reducing the number of non-perceptual errors that our current Go/NoGo design exploits.

### 3.2  PMC internal category signals and their functional roles

Our results add to a growing body of work showing that frontal cortex does not simply encode specific motor acts, but more abstract "change of behavior" signals. Recent electrophysiology in ferret frontal cortex [23] found that single neurons respond more strongly and persistently to sounds that instruct a change of ongoing behavior (Target cues) than to sounds that signal maintaining the current action (Reference cues), even when the same stimuli cue opposite movements and carry opposite reward valence across tasks. In line with this, PMC hemodynamic responses track the internal auditory category that drives a change in behavior: CBV patterns for difficult FAs resemble Hits, whereas easy FAs resemble CRs, indicating that PMC activity reflects the category the animal believes is present, rather than the eventual overt response. We note that, because fUSI relies on neurovascular coupling in much the same way as fMRI, these findings also have direct relevance for human cognitive neuroscience. Human imaging studies routinely decode choice, difficulty, and confidence from prefrontal and premotor BOLD signals [24,25], but typically treat all errors as a single category. Our data show that, at least in ferret PMC, hemodynamic activity primarily tracks the internal category (and associated change-of-behavior state) even when the overt report is wrong. This suggests that human studies could gain sensitivity and interpretability by explicitly manipulating and separating perceptual and non-perceptual errors (e.g., via difficulty and action-bias manipulations) instead of pooling them. Such approaches would be particularly appropriate in studies where higher levels of choice bias or exploratory choices are expected, as a lapse in attention may also involve a perceptual error depending on the lapse's timing.

### 3.3  Nature and neural origin of non-perceptual decision-making

Non-perceptual errors can arise from several sources: a bias toward a preferred choice, deliberate exploration, or lapses in attention that occur after the stimulus has been correctly perceived. Our Go/NoGo design was chosen precisely to create a strong preferred choice (the rewarded "Go"), thereby increasing the proportion of bias-driven non-perceptual errors. At the same time, internal states such as arousal, attention, reward anticipation, and action bias clearly influence whether the PMC categorical signal is ultimately translated into a lick or a withheld response. Work in macaque visual and PFC has identified a slow, brain-wide drift in population activity that covaries with Hit and FA rates and acts as an "impulsivity" or action bias signal, overriding largely stable sensory encoding at the moment of decision [26]. By analogy, we interpret non-perceptual errors in our task as cases where downstream circuits reading out PMC (possibly including basal ganglia [23]) combine a relatively stable representation of the inferred auditory category with fluctuating internal states, leading to inappropriate Go responses on some Reference trials despite correct PMC categorization. This may particularly be the case for the single "Restrained Licks", which appear reflexive rather than deliberate. Such reflexive actions likely engage circuits closely connected to the medullary reticular formation, the licking pattern generator (as shown in cats but likely also in the closely related ferret) [27]. A potential candidate region which would engage this formation is the orbitofrontal cortex (OFC). Being an agranular frontal cortex closely connected to limbic structures, OFC could influence reward-driven response criteria. Notably, stimulation of orbitofrontal regions in cats induces rhythmic licking [28]. Unfortunately, these areas were outside our imaging window. Future research should directly investigate these orbitofrontal and parietal regions to clarify their comparative roles in decision-making across species. In contrast, persistent licks may be more sensitive to the animal's broader motivational or exploratory state (e.g., frustration, thirst), or to history-dependent exploratory strategies in parietal cortex [29–31]. Our current data cannot cleanly decompose these contributions, but they suggest

that PMC provides a relatively stable internal category signal, while non-perceptual errors reflect how this signal is gated by state-dependent circuits in other frontal and parietal areas. Future work combining licking kinematics with independent markers of internal state (e.g., pupil size, heart rate) and targeted recordings in orbitofrontal and parietal cortices will be needed to disentangle action bias, exploration, and attentional lapses as distinct sources of non-perceptual errors.

## 4  Materials and methods

### 4.1  Ethics statement

Experiments were approved by the French Ministry of Agriculture (protocol authorization: 21022) and strictly comply with the European directives on the protection of animals used for scientific purposes (2010/63/EU).

### 4.2  Animal preparation

For the purposes of this study, two ferrets (ferrets A and B) were implanted with stainless steel headposts fixed to their skulls, in a procedure standard to the laboratory's prior work [19].

Following training, a frontal craniotomy was opened using a surgical micro-drill. Then, an ultrasound-transparent Poly-methylpentene (TPXTM) plate was placed above the brain and underneath the skull, secured with dental cement. In order to maintain the position of the ultrasound probe relative to the ferret during behavior, a custom-built stainless steel (440C) ferromagnetic frame (S8 Fig) was cemented to the ferret's implant surrounding the craniotomy, using a spirit level to keep it flat. The ultrasound probe could then be clamped to a paired magnetic frame, which could be magnetically attached to the ferret's implant. This system also allowed for the use of a microtranslator for the repositioning of the probe along the anterior-posterior dimension while maintaining the rigid attachment to the ferret's implant during their head-fixed behavior.

### 4.3  Stimuli and task

Both animals were trained on a Go/NoGo delayed response task. Each trial of the final task consisted of a 2-second Target or Reference AM pure tone, a 3-second post-stimulus silence, and the movement of the water spout in front of the ferret (Fig 1B). Licking the spout after a Target sound was rewarded with water. Licks after Reference sounds were discouraged with a 30-second time-out period before moving to the next trial. The valid lick-response window was 1 second for ferret A and 2 seconds for ferret B. An additional inter-trial interval (ITI) of 7 seconds was also enforced, irrespective of the ferret's choice. All stimuli were 65 dB.

Each ferret was trained to discriminate the AM tone stimuli along two different feature axes: carrier frequency or AM rate. During the "frequency rule" trials, all stimuli had a fixed AM rate of 5.8 Hz, and could have a carrier frequency selected from 521, 1,094, 2,514, 3,520, 8,089, or 17,000 Hz. During the "AM rule", all stimuli had a fixed carrier frequency of 5,657 Hz, and could have an AM rate selected from 1.2, 2.52, 4.8, 6.25, 11.9, or 25 Hz. Each ferret had to discriminate the three lowest AM rates or carrier frequencies from the three highest. Ferret A's Target stimuli were low, while ferret B's Target stimuli were high. Stimuli were chosen to be more than an octave apart from one another and to be easily discriminable across the category boundary, with the important exception of the two stimuli (per rule) close to the category boundary. These are called the "Difficult stimuli" as they were chosen to be more difficult to categorize. The remaining stimuli are called "Easy stimuli" due to the saturated performance levels. The choice of the distance between the difficult stimuli and the boundary was decided using the past behavioral results in [13], as the paper included data on behavioral performance for both AM rate discrimination and frequency discrimination.

Trials were organized into "feature-rule blocks". Within each block, ferrets could only be presented with sounds belonging to the prescribed rule. Blocks lasted 28 trials before switching; the 28 trials always consisted of 6 repetitions of each Easy stimulus and 2 repetitions of each Difficult stimulus, presented in pseudorandom order (with seed based on time). For every analysis, feature-rules are pooled together, as qualitatively the results within rules were indistinguishable (S6 Fig). The block structure was a vestige of an earlier training plan, and is irrelevant to all reported results.

Initial shaping was intended to train animals on a different final task, so some specifics in the stimuli were different. Ferrets were trained on two pairs of stimuli, 2,000 and 16,000 Hz tones modulated at 5.8 Hz for the frequency rule, and 5,657 Hz tones modulated at 2 and 17 Hz for the AM rule. Rules were swapped automatically every 50 trials. Initially, there was a 30 dB difference between Target (louder) and Reference stimuli. Ferrets took 1–2 weeks to demonstrate $d' > 1$ for this loudness-cued task, and would learn to perform the task more consistently and with no loudness cue in about 2 months. Ferrets were then trained on a task involving combinations of these stimuli for 2 months, after which the hypothesis for this paper was developed, and the strategy changed. Ferrets were then trained to tolerate the 3-second delay; ferret A took a month, ferret B less than a week. Ferrets were then presented with the new stimuli. Ferrets instantly generalized when presented with single pairs per rule per session, but did not when there were multiple pairs per rule; due to this, the delay was reduced and ferrets were retrained to stable performance with $d' > 1$, in the soundproof booth where the final recordings would take place. Following this, they learned in a few sessions, and the delay was gradually reintroduced over the course of two weeks. All data collection took place following these steps.

## 4.4 Behavior analyses

We used $d'$ to evaluate behavioral performance, a standard measure from signal detection theory [32,33]. It is calculated by subtracting the inverse-norm of the FA rate from the inverse-norm of the Hit rate. "Snooze" trials were excluded from all analyses. These were defined as sequences of consecutive trials for which the animal disengaged from the task. Because of the low criteria (as defined in signal detection theory) with which they performed the task (near-100% Hit rates), we used a highly conservative measure of snoozing, allowing for robust exclusion. We defined snooze blocks as any continuous sequence of trials for which the animals stopped licking that included at least 3 (missed) Go trials. If the ferret began to lick at stimuli again (Hit or FA), those trials and trials following them would be marked "non-snooze" unless the snooze criterion was again met. If the animal never licked until the end of the session, the last trial with a lick (Hit or FA) was the final non-snooze trial.

## 4.5 Preprocessing and fUSI analysis

### 4.5.1 fUSI recording and preprocessing.
Ultrasound Images were collected using an Iconeus One system (Iconeus, Paris, France), using the default settings and the IcoPrime probe (15 MHz central frequency, 70% bandwidth, 0.110 mm pitch, 128 elements). The Iconeus One System collects images at 500 Hz and applies a spatio-temporal clutter filter [34] to isolate blood vessels from other tissues using sets of 200 contiguous images. After filtering, the machine takes the mean power over these 200 frames, outputting vascular images at an effective framerate of 2.5 Hz. All timebin labels refer to the beginning of this composite image; thus for example, the frame labeled "4.6 s" includes fUSI data from 4.6 to 5.0 s.

### 4.5.2 Correction for brain deformations using NoRMCorre.
In fUSI, craniotomies are often quite large, and the brain is able to change shape along the course of a session. These movements tend to be slow, on the timescale of seconds. In order to correct for these movements, we applied the NoRMCorre algorithm [35,36]. NoRMCorre is a "piecewise rigid template alignment" algorithm, designed to correct for movements which are inconsistent between distant regions of an image but consistent in the near-field, and specifically designed with two photon imaging in mind. The algorithm takes a template image and splits it into "patches", each of which represents a smaller portion of the image. Every time frame is then compared to the template patches and adjusted to the position of peak cross-correlation. These adjusted patches are then recombined through interpolation.

The algorithm was designed for use in correcting for movements of cells in two-photon imaging. Its reliability, therefore, is dependent on consistent landmark signals being present at each time frame and each patch. While large blood vessels can serve this purpose for fUSI, they can be relatively sparse and are concentrated more visibly toward the surface of the brain. Patch sizes and overlaps were thus chosen conservatively, with a goal of correcting for vertical bulging movements of the brain. Specifically, every patch spanned the entire vertical range of each slice, and each was defined by a

600 micrometer central portion and 2,400 microns of overlap on either side of that portion horizontally (medio-laterally). Patches were selected from within a cropped area of each fUSI image containing all of the visible brain. These patches were adjusted vertically to counteract bulging movements that, when present, tended to be greater in the center of the craniotomy than toward the edge. Following the piecewise rigid template alignment, the patches were recombined using linear interpolation.

**4.5.3 ROI selection.** The anatomical positions of each fUSI recording were determined with the aid of a set of dedicated anatomical images, recorded on the first recording day following the implantation of the TPX. Coronal anatomical images of the entire craniotomy were taken with increments of 200 microns. The image depth was set to span the entire skull in order to aid in identifying landmarks. Each fUSI recording position was determined by comparing the vasculature of example images from each session to the dedicated anatomical images and to histological slices reported in [17]. This process is illustrated in more detail in S2 Fig.

This study focuses on the PMC. The PMC ROIs were manually selected within each recording using the same histological images as references, and each session's ROI was compared with a "template image" (one image per slice) to ensure similarity across sessions.

**4.5.4 Mean hemodynamic responses.** All reported functional results are based on the measure "%dCBV" or "Percent change in cortical blood volume". This is calculated for each time frame by subtracting the trial baseline mean of each voxel from the time frame, then dividing by the mean per voxel across all trials. The "baseline" time period was selected to be −1.8 to −0.6 seconds inclusive relative to stimulus onset. This %dCBV measure is calculated after all preprocessing steps are completed, and is used to calculate the reported means per voxel.

**4.5.5 Voxel averaging within columns.** This study additionally uses "columns" to organize the data for all major analyses (Fig 1F–1H). Data collected with fUSI shows strong correlations in responses over time between nearby pixels, in particular along the axis perpendicular to the gray matter surface. In order to better reflect this aspect of the data, increase signal-to-noise ratio, ease the pooling of data taken in different sessions, and decrease the number of variables inserted into later decoding models, voxels were placed into columns at fixed intervals along the cortical surface (Fig 1F), and the mean %dCBVs of each tile were analyzed directly. To do this, we used the "laynii" algorithm's [37,38] programs "LN_GROW_LAYERS" and "LN_COLUMNAR_DIST" to generate "columns" along the PMC after specifying the location of the surface and bottom of the ROI in a "rim" file and a seed column (on the medial edge of each ROI in this paper's case). The average column size created in this manner was 0.125 mm.

In order to accurately match the columns across sessions for the classifier models, sets of four contiguous tiles would be combined into bins of size 0.5 mm, in a manner such that these larger bins would approximately align across sessions (S5 Fig). In order for this to work, an anchor vessel was selected in each slice (by visual prominence), and a "origin" tile was set to fully contain the anchor vessel in its center. As long as the initial ROIs were similar before tile generation, this was sufficient such that additional landmark vessels would be located in the proper bins on inspection (S5 Fig). In order to verify this, all sessions were compared to a template set of bins for the session's slice.

## 4.6 Classifier analyses

**4.6.1 Training and evaluation of classifiers.** In order to assess similarities and differences in encoding of task variables in the PMC across conditions, we trained Linear Discriminant Analysis models. All classifiers were trained to discriminate between "Hit" trials and "CR" trials, where the number of trials in each category was counterbalanced by randomly selecting a number of "Hit" trials equal to the "CR" trials (as CR trials were always fewer). For each frame and slice (but across sessions), the classifier was trained using MATLAB's *fitcdiscr* function, with the columns' %dCBV values as features. We evaluated each classifier using 10-fold cross-validation, resulting in a "percent test-set correctly classified" for each condition. These measures are evaluated against a permutation test: the labels of each trial are shuffled 10,000 times to create a nonparametric noise distribution. Direct statistical judgments can then be made by comparing the

percent-correctly-classified against the (two-tailed) quantiles of the nonparametric distribution (Fig 1I). For all statistical tests performed on the later projection analyses, only frames for which the classification was above the 97.5th percentile were used. For the example stimulus-classifier plot, we underwent the same procedure for all pairs of Easy reference stimuli. The example plot shows the results from comparing the two frequency-rule tones at that slice; other slices and pairs showed similar results.

**4.6.2 Projection analyses.** Analyses of projections always used trials (FA trials or Difficult CR trials) which were not directly involved in the training of the Hit/CR models. We projected the trials onto the average classifier across folds. For convenience, we plotted the timecourses of these projections in example plots (in the manner of Fig 2B–2D) at all time frames in the trial, regardless of whether the classifier at that time-frame was significant. For the violin plots and statistical tests, we gathered summary statistics across time-frames (3.4–5.4 s post-stimulus window) and across slices. In order to do this, it was necessary to make the units of these classifiers comparable. To develop a method for this, the following reasoning was used: if a classifier is able to significantly distinguish between two categories, then the mean projected values of those two categories (Hit and CR in this case) may serve as reasonable anchors by which to align different axes to one another. To accomplish this, a scalar linear transformation was applied to each projected trial such that the mean projection of "Hit" trials would be 1 and the mean projection of "CR" trials would be −1. Specifically:

$$Proj_{out} = 1 - 2\frac{mean(Proj_{Hits}) - Proj_{in}}{(mean(Proj_{Hits}) - mean(Proj_{CRs}))} \tag{1}$$

where $Proj_{Hits}$ and $Proj_{CRs}$ are vectors of the projections of all (test) trials for a frame's classifier (Easy Hit and CR, respectively) and $Proj_{in}$ is whichever trial is currently being transformed (FA trials or Difficult CR trials). Projected trials with negative values would thus be closer to the mean CR projection, and projected trials with positive values would be closer to the mean Hit projection. It should again be stressed that this transformation can only be expected to be reasonable if the trained classifier is able to significantly distinguish the categories, otherwise the differences in mean projection for Hit and CR will be small and the remapping will be unwieldy.

In order to test differences between the mean projected values of various conditions, we elected to use a hierarchical bootstrap procedure [39]. Specifically, we simulated 100,000 bootstrapped replications of the experiment by, for each chosen slice location, randomly pulling (with replacement) sessions and trials, taking the mean across all trials. The number of sessions randomly chosen per slice was equivalent to the number of recorded sessions per slice, and the number of trials per session was equivalent to the number of trials for the chosen session. The 100,000 means collected in this manner were used to create the distributions presented in the violin plot for each conditions. In order to test differences between means, we followed the procedure suggested in [39]: we randomly paired bootstrapped means from each distribution to create a joint distribution, and tested to see which percentage of samples were on the appropriate side of the unit line (all tests were one-sided). Asterisks reflect one-sided tests: "*" means less than 4.55%, "**" means less than 0.27%, and "***" means less than 0.0063% (2, 3, and 4 standard deviations).

**4.6.3 Regression control for the influence of motor activity.** The ridge regression models used for the regression control were built from five variables: lick rate at four delays (1.2, 1.6, 2.0, and 2.4 seconds preceding the time frame) and motion energy at the moment of the recording frame. The ridge regression code used was *ridgeMML* from [40]. The manner in which lick rate was calculated will be discussed in the section "Video analysis of licking" below. Motion energy was calculated by taking the absolute difference between consecutive frames per pixel, and taking the mean across all pixels. Ridge regression models were built independently for each column and each frame, but across sessions for each column. All non-snooze trials were used to construct the models, agnostic of difficulty or response.

After building these models, we used them to predict the values of each column by first multiplying the regression weights by the regressor values for each trial and column (after subtracting the mean of each regressor across trials).

This reconstructs the mean column values recentered around the mean; therefore we re-added the means to get the final reconstructed values. These full predicted models were then subtracted from the actual recorded data, removing the components which could be explained by the ridge regression model. All classifier construction and projection steps were then repeated with the residual data.

### 4.7 Lick and motion analyses

**4.7.1 Video analysis of licking.** In order to evaluate the potential contributions of lick-evoked or lick-preparatory activity, lick rates had to be collected in the period before the spout arrived. To accomplish this, video data was recorded at a framerate of 7.5 Hz. Each frame was aligned to the fUSI data by using an LED set to light up each second starting at the beginning of each trial and ending before the ITI. Lick rate was calculated by drawing ROIs around the mouth of each ferret in each video (Fig 4A) and looking for changes in the color pink from frame to frame (in order to simply but effectively filter out whisker movements). This color difference was calculated by subtracting the blue signal from the red signal for each frame, then subtracting consecutive frames from one another and taking the absolute value. The threshold for making the lick judgments was set by eye per video, in order to give the best match to actual licks for the first 10 trials of each session when watching a video of the ferret. Each video frame was evaluated as having or not having a lick in a binary manner; these assessments were summed in order to downsample into the frame period of the fUSI acquisition (33 ms into 600 ms or 133 ms into 400 ms). For the violin plots in Fig 4B, we took the mean of this lick rate across frames 1–4.2 seconds for every trial and for each analyzed stimulus/response type.

**4.7.2 Determination of Persistent and Restrained licking.** Persistent and Restrained licking were defined in terms of licking behavior following the spout arrival. Licking intensity following spout arrival was quantified using a piezoelectric actuator attached to the water spout, which detected the mechanical vibrations caused by licks. Conceptually, Persistent licking was intended to capture continuous licking after an initial lick, either in a ferret's attempt to retrigger the piezo for an expected Hit, or because the ferret hoped to catch expected water. Persistent licking was thus defined as any trial where the piezoelectric device was activated for more than 30 ms in the period 200–400 ms following the response window, while all other FA trials were categorized as Restrained licking trials. This definition would also include other licking styles that extended into the later response window, such as a continuous licking unconcerned with the stimulus that may occur at the beginning of a session, but it was believed that in the case of perceptual errors the intended licking styles would occur often enough for the metric to capture them.

The threshold of 30 ms was chosen to capture all trials where one or multiple prolonged licks occurred in the 200–400 ms response window. Due to the piezoelectric device's inherent mechanical ringing properties, it exhibited rapid and intermittent on/off patterns even following single licks. To accurately capture periods of continuous licking, piezo activations occurring within a 37 ms interval were considered as part of a single lick bout and time frames in between were flagged as "activated".

## Supporting information

**S1 Fig. Behavior plots for both rules. A**: Cartoon showing the organization of the AM pure tones in each rule for ferret B. "T" is a go stimulus, "R" is a reference stimulus. Ferret A would be the same, but with T and R locations reversed. **B**: Measured psychometric curves for ferret A, for each rule. **C**: Same as B, but for ferret B.
(EPS)

**S2 Fig. Demonstration of the method used for judging anatomical correspondences.** In order to judge the anatomical location of any particular micromanipulator position, information from all anatomical images was taken into account. The images for each ferret were taken at anterio-posterior steps of 200 micrometers. Images on the left are anterior, and on the right are posterior. The AP position estimates can be used for comparison to pages 86, 92, 98, 104, and 110 of the

Radtke-Schuller ferret atlas [17]. Particularly important landmarks in this case include the cruciate sulcus, which branches out from the interhemispheric fissure in a Y formation. Moving anterior from the branch point, it becomes more and more lateral. Also important and reliable as an anatomical anchor is the corpus callosum: when it is present, the large blood vessel in the intrahemispheric fissure has a "gap" in the image. Moving anterior, the corpus callosum is no longer present. This transition, according to the atlas, happens in between 26.1 and 25.5 mm anterior to the occipital crest. The example slice in the text was ferret A at −25.9. Green boxes surround the locations used for repeated recordings; an arrow at the bottom points to the recording location used as an example throughout the text.
(EPS)

**S3 Fig. Replication of example figures from Figs 1, 2, and 3 for all slices and both ferrets.** The ferret and estimated slice AP position are noted on the left of each row. The example slice from the main text is in the second row. **A, E, I, M**: Percents correctly classified for each slice, as in Fig 1I. **B, F, J, N**: Projection plots as in Fig 2. Easy FA, Difficult FA, and Difficult CR projections are plotted on the same figure to save space. In addition, Difficult Hit and Easy Miss projections are included. **C, G, K, O**: Projection plots for Persistent/Restrained Lick trials among Difficult FA trials, as in Fig 3A. **D, H, L, P**: Projection plots for Persistent/Restrained Lick trials among Easy FA trials, as in Fig 3B.
(EPS)

**S4 Fig. Replication of example figures from Fig 4 for all slices and both ferrets.** Same as S3 Fig, except after subtracting the regression model, as in Fig 4.
(EPS)

**S5 Fig. Consistency of anatomy and columns across recording sessions.** Three example sessions are chosen for each ferret, using the same slice; the recordings were each taken approximately a week apart. Despite this, the gross anatomy looks the same for each image. Column positions were placed such that the positions of major vascular landmarks would be the same relative to each column; see, for example, the large vertical vessel under position "0" in ferret A, and the large vessel between columns 1.5 and 2.
(EPS)

**S6 Fig. Mean responses for all stimuli, regions, and ferrets. A, C, E, G, I**: Mean Hit responses for each recording location for each ferret. Each of the six individual plots in a figure is a single stimulus; columns designate the distance from the boundary, while rows indicate the AM/freq rule. **B, D, F, H, J**: Same, but for CRs instead of Hits.
(EPS)

**S7 Fig. Persistent licks are reduced on Easy false alarm trials. A**: Violin plots of the proportion of persistent licks in False Alarm trials across sessions for ferret A as a function of difficulty level. Each dot represents one session. Sessions with missing values or zero rates in either condition were excluded. **B**: Same as **A** for ferret B. **C**: Same as A for Hit trials (ferret A). **D**: Same as A for Hit trials (ferret B). A paired $t$ test was performed within each panel to compare difficult vs. easy trials, and significance is indicated above the violins ($p < 0.05$ *, $p < 0.01$ **, $p < 0.001$ ***; n.s., not significant).
(EPS)

**S8 Fig. Design of the magnetic frame system for stabilizing the fUSI recordings against movements. A**: Top view of the ferromagnetic frame. The frame was made of the corrosion-resistant, ferromagnetic stainless steel 440C. **B**: Bottom view of the ferromagnetic frame. The legs were used to aid in cementing by steadying the frame in place and providing locations to cement. **C**: Top view of the magnetic frame. The frame was made of aluminum, and designed to hold the micromanipulator and supermagnets while otherwise minimizing the space taken up. **D**: Bottom view of the magnetic frame. The supermagnets were stored in the three holes on the bottom; a later version (blueprint in Fig H) would have five holes for magnets. **E**: Photo of magnetic frame with micromanipulator and the probe-clamp. **F**: Photo of the 5-peg

ferromagnetic frame. **G, H**: Detailed blueprints for the ferromagnetic frame (G) and magnetic frame (H) given to the metal shop which constructed them.
(EPS)

## Acknowledgments

We would like to thank Thibaut Doisy, Denis Lancelin, Gwenola Dubois, Balkis Cadi, and Lynda Bourguignon for technical and administrative support throughout the project.

## Author contributions

**Conceptualization:** Jeff Boucher, Yves Boubenec.

**Data curation:** Jeff Boucher.

**Formal analysis:** Jeff Boucher.

**Funding acquisition:** Jeff Boucher, Shihab Shamma, Yves Boubenec.

**Investigation:** Jeff Boucher, Yves Boubenec.

**Methodology:** Jeff Boucher, Yves Boubenec.

**Project administration:** Yves Boubenec.

**Resources:** Yves Boubenec.

**Software:** Jeff Boucher, Yves Boubenec.

**Supervision:** Shihab Shamma, Yves Boubenec.

**Validation:** Jeff Boucher, Yves Boubenec.

**Visualization:** Jeff Boucher, Yves Boubenec.

**Writing – original draft:** Jeff Boucher.

**Writing – review & editing:** Jeff Boucher, Shihab Shamma, Yves Boubenec.

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
