## [Editor Report · Decision Letter 0]

15 May 2025

Dear Dr Boucher,

Thank you for submitting your manuscript entitled "Premotor cortex reflects internal auditory category, not reported category" for consideration as a Short Reports by PLOS Biology.

Your manuscript has now been evaluated by the PLOS Biology editorial staff, as well as by an academic editor with relevant expertise, and I am writing to let you know that we would like to send your submission out for external peer review.

However, before we can send your manuscript to reviewers, we need you to complete your submission by providing the metadata that is required for full assessment. To this end, please login to Editorial Manager where you will find the paper in the ‘Submissions Needing Revisions’ folder on your homepage. Please click ‘Revise Submission’ from the Action Links and complete all additional questions in the submission questionnaire.

Once your full submission is complete, your paper will undergo a series of checks in preparation for peer review. After your manuscript has passed the checks it will be sent out for review. To provide the metadata for your submission, please Login to Editorial Manager (https://www.editorialmanager.com/pbiology) within two working days, i.e. by May 17 2025 11:59PM.

If you would like us to consider previous reviewer reports, please edit your cover letter to let us know and include the name of the journal where the work was previously considered and the manuscript ID it was given. In addition, please upload a response to the reviews as a ‘Prior Peer Review’ file type, which should include the reports in full and a point-by-point reply detailing how you have or plan to address the reviewers’ concerns.

Kind regards,

Taylor

Taylor Hart, PhD,

Associate Editor

PLOS Biology

thart@plos.org

---

## [Decision Letter · Decision Letter 1]

23 Jul 2025

Dear Dr Boucher,

Thank you for your patience while your manuscript "Premotor cortex reflects internal auditory category, not reported category" was peer-reviewed at PLOS Biology. It has now been evaluated by the PLOS Biology editors, an Academic Editor with relevant expertise, and by several independent reviewers.

In light of the reviews, which you will find at the end of this email, we would like to invite you to revise the work to thoroughly address the reviewers' reports.

The reviewers say that the results are novel and compelling. However, they noted areas requiring additional support or clarifications related to the study's framing and discussion, the task design, and the data presentation. One reviewer raised a concern about the level of statistical power. We invite you to substantially revise your study in response to these points, including by performing new analyses, providing the missing details and data, and through textual revisions.

Given the extent of revision needed, we cannot make a decision about publication until we have seen the revised manuscript and your response to the reviewers' comments. Your revised manuscript is likely to be sent for further evaluation by all or a subset of the reviewers.

**IMPORTANT - SUBMITTING YOUR REVISION**

*Re-submission Checklist*

*Published Peer Review*

*PLOS Data Policy*

*Blot and Gel Data Policy*

Sincerely,

Taylor

Taylor Hart, PhD,

Associate Editor

PLOS Biology

thart@plos.org

REVIEWS:

Reviewer #1: This manuscript examines how stimuli are represented in the premotor cortex (PMC) of ferrets during decision-making, using functional ultrasound (fUS) imaging. The authors use a Go/NoGo delayed categorization task to disentangle whether premotor signals reflect stimulus perception or the eventual behavioral output. In the task, pure tone stimuli were categorized as 'Target' (requiring a lick response) or 'Reference' (requiring the animal to withhold licking). Task difficulty was manipulated by varying the distance of tones from the category boundary—tones farther from the boundary created easier trials, while those near the boundary created more difficult trials.

This gradation in difficulty was used to interpret error trials: errors on easy trials were assumed to be non-perceptual (i.e., due to lapses), while errors on difficult trials were interpreted as perceptual. fUS signals could robustly distinguish between easy Hit trials and Correct Rejections (CR), due to distinct cerebral blood volume (CBV) changes. These classifier axes were then used to project error trials. Easy errors were more similar to CRs, while difficult errors resembled Hits, suggesting that PMC activity reflects stimulus category representation rather than simply motor output. Licking behaviors, including "restrained" versus "persistent" licking, were carefully analyzed to support this interpretation.

Overall, this is an intriguing and well-motivated study. It leverages new neuroimaging technology in an animal model that is less commonly used, and the behavioral and neural analyses are thoughtfully structured. Below are comments to help strengthen the interpretation of the results.

Major Comments

Behavioral Training Details

The behavioral training description (e.g., Line 236) is unclear. Were the initial shaping sessions part of data collection? How long did it take to train the animals to criterion? What steps were involved? Also, there appears to be a notable number of false alarms even in the 'easy' stimulus region—was this expected? A 30-second timeout seems long; did this influence behavior or motivation?

fUS Analysis Across Rule Types

Were the 'feature-rule' and 'amplitude-rule' blocks analyzed separately or pooled? If they were pooled, could this mask rule-specific effects? A separation might provide insight into whether PMC activity generalizes across task contexts.

Voxel Averaging and Classifier Training

The approach of averaging voxels into columns for session pooling is an interesting strategy. However, it is unclear whether classifiers were trained per animal or on combined data. If classifiers were trained separately, does the column-averaging allow for cross-animal data integration? Clarifying this could help interpret the generalizability of the classifiers.

PMC-Auditory Cortex Relationship

There is apparent stimulus-related classification accuracy shortly after tone onset (e.g., Figures 1I, 4C, and Supplementary Figure 6). Could the authors elaborate on potential auditory processing in PMC or rapid top-down feedback? Is there known auditory input to PMC in ferrets?

CBV Decreases During Hits

What is the interpretation of the negative CBV deflections during Hit trials? Are similar spatial patterns observed during difficult trials (analogous to Figure 1E/G/H)? It would be helpful to directly compare CBV maps for Hits and CRs across difficulty levels.

Licking Behavior Dynamics

Do restrained versus persistent licking behaviors vary across the session timecourse or depend on trial type (e.g., more persistent licking during Target trials or at later session stages)? This could influence interpretation of motor contributions to the signal.

Minor Comments

Define FA in Context

Line 45: Please define "FA trial" (false alarm) in the context of signal detection theory. This would help clarify the expected neural similarity to CR trials.

Clarify 'Similar' in Line 54

Line 54: What aspect of Easy and Difficult CRs were 'similar'? CBV patterns? Behavioral outcomes? Please specify.

Report Results for Difficult Targets

Line 69: For completeness, include the results from difficult Target trials, as these could inform the perceptual decision-making analysis.

Clarify Figure 1 Legend

The legend for Figure 1 is unclear. Does "example pair of two reference stimuli" refer to the two 'easy' reference stimuli from the two-frequency rule?

Typo Correction

Line 292: "Vowel averaging" should be "voxel averaging." Please correct this typographical error.

Reviewer #2: This study investigates the role of the premotor cortex (PMC) in auditory categorization using functional ultrasound imaging in ferrets performing a Go/No-Go delayed categorization task. It provides novel and valuable insights into the types of signals encoded in the PMC—choice versus perception—in response to task stimuli. This information is relevant to perceptual decision-making research, as it challenges assumptions that all behavioral task errors are purely perceptual, or that errors in perceptual tasks are idiosyncratic and uninterpretable. The clear differentiation of error types represents a novel contribution. While the experimental design and methodology are generally sound, additional feedback from researchers familiar with the functional ultrasound imaging technique could strengthen the manuscript.

The design of the behavioral task, combined with neural activity recordings, effectively demonstrates that errors occurring with stimuli closer to the category boundary (Difficult-Ref) are differentially encoded in the PMC compared to errors occurring with stimuli further from the boundary (Easy-Ref). Specifically, Difficult-Ref FAs are more similar to Hit trials, while Easy-Ref FAs resemble Correct Rejections.

However, some aspects of task design remain unclear, potentially leading to alternative interpretations of the main conclusion regarding representation of "subjective stimulus category." Incomplete data presentation further complicates full evaluation and interpretation of the claims. Enhanced descriptions of experiments and clearer result presentation would better support the conclusions.

Major:

1.Task design and reward expectation: Why is reward provided only for Hit and not CR trials, creating an asymmetric reward schedule and reward expectation scenario? Only licking and targets yield rewards, making perceptual category representation potentially confounded with reward expectation. Could the authors clarify the reasoning behind this task design? How can reward expectation and sensory processing be distinguished? Why is the task described as detection rather than discrimination?

2.Interpretation of FAs: The authors claim Easy-Ref false alarms mainly reflect non-perceptual errors, whereas Difficult-Ref false alarms reflect perceptual errors. This claim appears overstated, considering that the FA rate for Easy-Ref trials is high (~50% for A, ~60% for B; Fig. 1C), and Difficult-Ref false alarms are about ~10 points higher. Generative models would thus predict that most Difficult-Ref FAs remain non-perceptual. Please clarify the reasoning behind these statements and provide estimates of perceptual FAs for each category. Is additional stratification of Difficult-Ref FAs possible?

3.Psychometric curve: Descriptions of Difficult-Ref and Easy-Ref trials use terms such as "slope" and "saturated," which are not clearly defined. Provide a quantitative description and clear definition of slope, saturation, and trial types (Ref, Target, difficulty levels) within the psychometric function.

4.Baseline differences between groups: In Fig. 1H, the data appear similar to Fig. 1G but shifted back by 1-2 seconds. Could the authors clarify whether this is intentional or an error, and explain the presence of inhibition specifically before sound presentation in Correct Rejection trials?

Minor:

5.Behavioral task: Please provide clearer explanations of the two task versions (AM vs. FM rule) and clarify the "block logic." Although the methods state similar results across task versions, psychometric plots for each task type and animal would ne important (as is standard in the field). Clearly define parameters determining Easy-Ref and Difficult-Ref trials.

6.Include more details of task stimuli and organization within the main text (currently only available in supplementary data), specifying clearly what AM and FM refer to, block organization, and definitions of Easy-Ref and Difficult-Ref stimuli (dB and Hz). Clarify differences between ferrets 1 and 2 in terms of target/reference stimuli in a cohesive section.

7.Regarding licking, clarify the validity of binarizing each trial into "persistent" vs. "restrained" (piezoelectric sensor activation >30 ms during 200-400 ms after the response window). E.g., provide histograms or similar visualizations to validate the approach. Is it indeed the case (as needed for the argument), that persistent licking between Easy-Ref and Difficult-Ref FAs is similar (e.g., lick counts, bouts)?

8.Provide additional explanation of PMC columns recorded are important: Could the authors clarify whether there are activity differences within PMC by depth, especially since Supplementary figures for ferret 2 differ from main text figures for ferret 1. Please justify presenting data from only one animal and single depth in the main text.

9.No data shown for miss trials: How does PMC activity during Miss trials appear, especially within the Hit/CR encoding space used in Fig. 2? Plot Miss trials for comparison with Hit, CR, and FA trials.

10.Is there evidence supporting the idea that additional waiting during reference stimuli represents a "punishment" (as stated) for ferrets?

11.In line 75, clarify if the stated n-values refer to total Difficult-Ref or Easy-Ref trials, or specifically to False Alarms within those categories.

12.The title appears somewhat confusing, as the abstract explains that PMC encodes both real and reported auditory categories, depending on trial difficulty.

13.Remove the comma in the abstract's final phrase: "motivational information."

14.Clearly define "False Alarms" and "AM" upon first mention.

15.Differentiate colors for Difficult-Ref and Easy-Ref FAs from the persistent vs. restrained licking categories in Supplementary Figure 4C-F. There is currently inconsistent color coding and figure layouts across these figures.

Reviewer #3 [Grace Ross]: High level summary: In this manuscript, the authors investigate whether neural activity in the ferret PMC - measured by hemodynamic response - reflects stimulus categorization or action in perceptual decision making during a head-fixed perceptual decision auditory task. The authors present compelling evidence that the hemodynamic responses in PMC differ based on perceptual difficulty and trial outcome, however additional analysis is needed to fully understand how PMC hemodynamic response differs in Easy and Difficult trials and an expanded discussion on how perceptual vs non-perceptual errors are classified in this task design is necessary to clarify the paper's conclusions. The argument that Easy task is more prone to non-perceptual errors is very interesting, and it make sense that the reason for an error in an Easy vs Difficult task is different, however the authors don't provide thorough behavioral evidence or interpretation to be convincing what the source of non-perceptual errors are.

INTRODUCTION

Comment: I would consider a visual representation of the tested hypothesis with the experimental design, it gets quite confusing to think through the different conditions and what is predicted. A diagram would make this easier for the reader - maybe an addition to Figure 1?

Comment: When writing the hypothesis earlier in the introduction, clearly refer to it as the 'stimulus category hypothesis' or reword in line 56, with the name in quotes I feel like it's referring back to something that was defined earlier in the text and interrupts flow

Comment: Lines 54 to 58 I would be precise when making comparisons here, when the text says something like 'FA errors in Easy trials were also similar to CR trials…." it's not clear what similarity you're referring to, if you're talking about number of FA and CR, or the hemodynamic response. It does become clear in the next couple of sentences that you're discussing the hemodynamic response, but upon first impression it is unclear. Please edit language to be more precise.

Comment: Introduction What is known about the primate/rodent/human prefrontal response to perceptual versus non-perceptual errors? I would contextualize open questions in the ferret with other model organisms and humans to further demonstrate/detail the need for this study. I also think that when working with a less common model organism like the ferret, context is important and can add a lot of relevance to your manuscript.

RESULTS

Comment: Line 72 Please list stats for difficult stimuli as with easy stimuli

Comment: are there enough non correct outcomes in the easy trials to be powered? Please provide a comprehensive breakdown of performance percentage's of each trial outcome for each condition, for both animals.

Comment: Figure 1E - confused how accuracy can be predicted along the entire time axis? I think this method needs more explanation of what the graph is actually showing….I'm still not sure what 'latency' is being referred to in line 94 and how what is in the graph relates to the hemodynamic response?

Comment: Figure 1E: is there a significance test that can be done at time -1 between Hit and CR? Or I would at least acknowledge in the text that even before the stimuli, there is a visible difference between Hit and CR, why that is and what that could mean

Comment: [Sec sec011] How has the relationship between perceptual and non-perceptual errors and the Easy and Difficult task conditions been shown? Not controlling for a marker of attention or motivation, am I missing something or is this an assumption that's being made? I think explicit evidence is needed for this statement to hold more weight, especially as it is a primary part of this manuscript

Comment Fig2 B: If easy FA and easy CR look similar, but easy hit looks quite different, then the argument is that the animal can detect the reference stimuli but chooses sometimes to respond and other times not to. But when the target stimuli is present, this response is much higher because it's reward anticipatory? What is the argument that the animal is gaining by responding to the reference stimuli even though it's properly classified?

Comment Fig2C: the animal's PMC response is similar for a correct target trial and a difficult FA so the argument is that the animal thinks the tone they're responding to is correct because it matches to the Easy hit. Why isn't the Difficult FA and Difficult Hit plotted together? I would like to see that, as I think it's more informative because the comparison between an easy hit and difficult FA is not direct because the experimental parameters are different. I guess this is in E, but the metric is different so it doesn't feel like a clear comparison? I would still plot it and see if it's informative

Comment Fig2D: A difficult CR where the animal identifies the reference stimuli and abstains is the same as in the Easy task. So the argument is that regardless of task condition, the response from PMC when inhibiting a response to the non-target stimuli is the same. Does this make sense, in terms of the role of prefrontal regions in responding differentially in task difficulty comparisons? Maybe that's dlPFC, but in PMC, I would discuss what this means because I'm surprised by this result - but maybe I'm misunderstanding

Comment Fig2E: from 3.4 to 5.4 seconds post stimulus onset. So a difficult FA elicits the strongest PMC response compared to a difficult CR and an Easy FA. I want a comparison of Difficult FA, Difficult CR and Difficult Hit. And if a false alarm is eliciting the largest reaction in PMC hemodynamics - if I'm understanding this graph correctly - then that should be thoroughly discussed. I'm not sure I agree with the choice of these comparisons. Fig2B needs to be replicated for Difficult trials. And looks like the ramp up of activity is only in trial outcomes that elicit a response, so more of an anticipatory signal than anything else?

Comment: 3.4 Is an anticipatory response considered perceptual or non-perceptual?

DISCUSSION

Comment: 4.1 I would strengthen the translational relevance of what the fUS results add to our knowledge as a field and how the addition of ephys would further this understanding. Instead of simply stating that ephys investigations are needed, I would hypothesize what those results may look like based on what is known about the electrophysiological activity of the ferret frontal cortex as a whole and specifically PFC vs PMC. This section reads too similar to a limitation section without a broader discussion of (1) what specifically does fUS add to current understanding/why it was chosen over ephys, (2) what do you predict based on the literature to see from an ephys investigation and how those results would meaningfully contribute to current knowledge.

Comment: 4.2 I think discussing other internal states such as attention and reward anticipation are important to this discussion section. There was no discussion of impulsivity prior and it feels a bit out of the blue. How does this relate to perceptual vs non-perceptual decision making? This may be a good place to discuss how you might begin to disentangle the sources of non-perceptual errors, which I think is missing

Comment 4.3 Propose downstream circuits override PMC categorical signals, causing non-perceptual errors. I would orient the reader a bit more to this discussion section with an example following the first sentence. Why would restrained licks be reflexive rather than deliberate? I would have thought the opposite.

Comment: I would add a final discussion section about the impact of this research on the ferret neuroscience community, the prefrontal cortex community and - what I think is most missing - the human cognitive neuroscience community. These results are very interesting, what I think is missing is additional contextualization and answering the 'why does this matter' question a lot of readers may have.

Summary of specific figure requests/suggestions:

-Visual representation of hypotheses to improve readability

-I would like to see simple bar graphs of each ferret's performance in Easy and Difficult sessions, with percent of each trial outcome

-Fig 2B plot replicated for Difficult trials

---

## [Decision Letter · Decision Letter 2]

19 Mar 2026

Dear Dr Boucher,

Thank you for your patience while we considered your revised manuscript "Premotor cortex reflects internal auditory category, not reported category" for publication as a Short Reports at PLOS Biology. This revised version of your manuscript has been evaluated by the PLOS Biology editors, the Academic Editor, and the original reviewers.

Based on the reviews, we are likely to accept this manuscript for publication. Please also make sure to address the following data and other policy-related requests.

IMPORTANT: Please ensure that your next revision addresses all of the following points:

**Title:

We think that your paper's title could be made more broadly accessible by explaining a bit more of the specific context. We propose two alternative versions that we think would achieve this. Is either of these options acceptable to you?

"Premotor cortex hemodynamic responses primarily reflect perceptual rather than specific motor aspects of decision making."

Or

"Premotor cortex integrates auditory inputs to guide behavior based on perceptual information but not motivational information"

**Ethics:

Please move the information about ethical compliance to a new, first subheading of the Materials and Methods section, titled "Ethics statement".

**Data:

Note that we do not require the raw imaging data (although we encourage you to upload this to a publicly available database), but we do require the numerical values underlying the plots. Please supply the numerical values either in a supplementary excel file or as a permanent DOI’d deposition (not Github alone) for the following figure panels:

2BCDE

3CDE

4BDEFGH

Please ensure that your Data Statement in the submission system accurately describes where your data can be found and is in final format, as it will be published as written there.

Please also cite the location of the data clearly in all relevant main and supplementary Figure legends, e.g. “The data underlying this Figure can be found in S1 Data” or “The data underlying this Figure can be found in https://doi.org/10.5281/zenodo.XXXXX”

**Supplement:

Please include the titles and legends for all supplementary items in the main manuscript file. You can then remove the supplementary PDF, as long as all supplementary figures are also uploaded separately.

**Code:

Per journal policy, if you have generated any custom code or scripts during the course of this investigation, we require that you make it available without restrictions. Please ensure that the code is sufficiently well documented and reusable, and that your Data Statement in the Editorial Manager submission system accurately describes where your code can be found. Please also ensure that you choose a license for your code and include a Readme file.

We expect to receive your revised manuscript within two weeks.

*Published Peer Review History*

*Press*

Sincerely,

Taylor

Taylor Hart, PhD,

Associate Editor

thart@plos.org

PLOS Biology

REVIEWS

Reviewer #1:

The authors have satisfactorily addressed my comments. The expansion of the methods and task design, alongside additional analyses and more nuanced interpretation, strengthen the manuscript and are very helpful in interpreting the interesting results.

Reviewer #2: The authors have comprehensively addressed my concerns.

Reviewer #3: [This reviewer recommended to Accept without any further comment]

---

## [Editor Report · Decision Letter 3]

7 Apr 2026

Dear Dr Boucher,

Thank you for the submission of your revised Short Report "Premotor cortex hemodynamic responses primarily reflect perceptual rather than specific motor aspects of decision making" for publication in PLOS Biology. On behalf of my colleagues and the Academic Editor, Thomas Klausberger, I am pleased to say that we can in principle accept your manuscript for publication, provided you address any remaining formatting and reporting issues. These will be detailed in an email you should receive within 2-3 business days from our colleagues in the journal operations team; no action is required from you until then. Please note that we will not be able to formally accept your manuscript and schedule it for publication until you have completed any requested changes.

PRESS

Sincerely,

Taylor

Taylor Hart, PhD,

Associate Editor

PLOS Biology

thart@plos.org